# Influence of Phases of Coherence Functions on the Wind Field Simulation Using Spectral Representation Method

**Ning Zhao [1], Xiaolong Li [1], Liuliu Peng [2,\*], Zhilong Xu [1], Xiaowei Chen [1] and Xuewei Wang [1,\*]**

1   School of Civil Engineering, Sichuan Agricultural University, Dujiangyan 611830, China
2   School of Civil Engineering, Chongqing University, Chongqing 400044, China
\*   Correspondence: liuliu.peng@cqu.edu.cn (L.P.); wxw5778@163.com (X.W.)

**Abstract:** The wave passage effect is a measure of the wave passage delay due to the apparent velocity of waves, which is one of spatially varying properties of multivariate random processes. The phase of coherence function reflects the wave passage effect of wind fields. In the wind field, simulation by the spectral representation method, the classical phase formula, is not rigorous. This may affect the accuracy of simulation results and even cause incorrect simulations. In this study, the influences of the phase on stationary and nonstationary wind field simulations are researched and discussed in detail. Two schemes containing the classical phase formula and the separated phase scheme are compared in four types of wind field simulation. The qualitative analysis based on theoretical correlation function formula is first made to study the influence of the phase. Then, four numerical examples are utilized to quantitatively study the magnitude of the influence on the sample time history and correlation function of the simulated wind field. Results show that the classical phase formula will result in considerable simulation error for all four types of wind fields because it cannot completely represent the phase angle of a complex number.

**Keywords:** wind field; spectral representation method; simulation; phase; error

## 1. Introduction

With the rapid development of computer technology, Monte Carlo simulation is widely used in the field of engineering to solve complex problems, such as nonlinearity, system randomness, and stochastic stability [1,2]. An important part of the Monte Carlo simulation is to simulate the samples of random excitations, such as wind, ground motion, and wave [3]. Wind field and ground motion are usually modeled as random processes, while wave field is usually modeled as random field or random wave. Depending on the situation, the above simulations can be classified as one-dimensional or multidimensional [1,4], single or multi-point [5,6], ergodic or non-ergodic [6,7], stationary or nonstationary [8–11], or Gaussian or non-Gaussian [12–16]. In this study, the wind field is taken as an example to research the simulation of one-dimensional, multi-point, and Gaussian random processes.

The one-dimensional, multi-point, and Gaussian random wind field can be classified as four types: non-ergodic stationary wind field, ergodic stationary wind field, time-invariant coherent nonstationary wind field, and time-varying coherent nonstationary wind field, based on the probabilistic characteristics. The task of wind field simulation is to generate sample functions having prescribed probabilistic characteristics. In wind field simulation, the spectral representation method is more widely used than other approaches, such as the time series method and time-frequency analysis method, because of its accuracy and simplicity. In recent decades, researchers have done a lot of work to optimize and improve the spectral representation method in wind field simulation.

In the simulation of the non-ergodic stationary wind field, Yang et al. [7] proposed a closed-form solution of Cholesky decomposition to simplify the spectral matrix decomposition. Chen et al. [17] introduced an improved fast Fourier transform (FFT) method to

simulate the short-term wind velocity field. In addition, some methods based on stochastic wave were developed [18,19]. In terms of the ergodic stationary wind field, Cao et al. [20] introduced the closed-form solution of Cholesky decomposition. Togbenou et al. [21] proposed an efficient simulation method for vertically distributed wind velocity field, based on approximate piecewise wind spectrum. Zhao and Huang [22] proposed an enhanced closed-form solution of Cholesky decomposition for the wind field along an arbitrary axis. Zhao et al. [23] established an enhanced spectral representation method with high accuracy and efficiency for the ergodic wind field simulation. Additionally, the interpolation schemes were also developed to optimize the spectral matrix decomposition [24,25]. For the time-invariant coherent nonstationary wind field, the spectral matrix decomposition is the same as the stationary simulation, and thus most studies focus on the introduction and optimization of FFT [26–29]. Besides, the stochastic wave approach is also introduced to the nonstationary simulation by proper orthogonal decomposition [30]. The simulation of the time-varying coherent nonstationary wind field is optimized and enhanced mainly based on the interpolation and time-frequency decoupling techniques [31–34].

However, the classical phase formula in almost all the above-mentioned simulation methods is not rigorous. This may cause misunderstanding for the wind field simulation, considering the wave passage effect. In this study, the influence of the classical phase formula on the simulation of stationary and nonstationary wind fields is analyzed and discussed in depth. Firstly, the theory of the wind field simulation based on spectral representation method are reviewed. Secondly, two treatment schemes of deterministic phases in the simulation formula are introduced. Then, the influence of phase on correlation functions in four types of the wind field simulation is qualitatively discussed, based on the theoretical formula. Further, numerical examples of four types of the wind field simulation are utilized to conduct a quantitative analysis about the influence of phase on samples and correlation functions. Finally, some conclusions are drawn.

## 2. Wind Field Simulation by Spectral Representation Method

Based on the different probabilistic properties of a given wind field, the wind speed field can be described as the following four types: (1) stationary non-ergodic wind field, (2) stationary ergodic wind field, (3) nonstationary wind field with time-invariant coherence function, and (4) nonstationary wind field with time-varying coherence function. The first two are stationary wind fields and the last two are nonstationary wind fields. The spectral representation simulation process of these four wind fields will be, respectively, introduced below. In order to clearly reflect the phases of coherence functions, the SRM based on the decomposition of coherence matrix [35,36] will be presented.

### 2.1. Stationary Wind Fields

Consider an $n$-variate vector-value stationary wind fluctuation process $\mathbf{x}(t) = [x_1(t), x_2(t), \cdots, x_n(t)]^T$, where $T$ denotes transpose. Its spectral matrix is given by

$$\mathbf{S}(\omega) = \begin{bmatrix} S_{11}(\omega) & S_{12}(\omega) & \cdots & S_{1n}(\omega) \\ S_{21}(\omega) & S_{22}(\omega) & \cdots & S_{2n}(\omega) \\ \vdots & \vdots & \ddots & \vdots \\ S_{n1}(\omega) & S_{n2}(\omega) & \cdots & S_{nn}(\omega) \end{bmatrix} \tag{1}$$

$$S_{jk}(\omega, t) = \sqrt{S_{jj}(\omega, t) S_{kk}(\omega, t)} \gamma_{jk}(\omega) \tag{2}$$

where $\mathbf{S}(\omega)$ = a Hermitian matrix with positive definite property; $\omega$ = circle frequency; $S_{jj}(\omega), j = 1, 2, \cdots, n$ = auto-PSD function of $x_j(t)$ and a real and non-negative function of $\omega$; $S_{jk}(\omega), j = 1, 2, \cdots, n; j \neq k$ = cross-PSD function between $x_j(t)$ and $x_k(t)$ and a complex

function of $\omega$; $\gamma_{jk}(\omega)$ = coherence function between $x_j(t)$ and $x_k(t)$. In order to consider the wave passage effect, $\gamma_{jk}(\omega)$ should be a complex number and can be expressed as

$$\gamma_{jk}(\omega) = \left|\gamma_{jk}(\omega)\right|e^{i\vartheta_{jk}(\omega)} \tag{3}$$

where $i = \sqrt{-1}$ = imaginary unit; $|\cdot|$ = modulus of a complex number; $\vartheta_{jk}(\omega)$ = complex phase of $\gamma_{jk}(\omega)$.

According to Equations (1) and (2), the spectral matrix can be represented as

$$\mathbf{S}(\omega) = \mathbf{D}(\omega)\mathbf{\Gamma}(\omega)\mathbf{D}(\omega) \tag{4}$$

where $\mathbf{D}(\omega)$ is a diagonal matrix composed of auto-PSDs, i.e.,

$$\mathbf{D}(\omega) = diag\left[\sqrt{S_{11}(\omega)}, \sqrt{S_{22}(\omega)}, \cdots, \sqrt{S_{nn}(\omega)}\right] \tag{5}$$

and $\mathbf{\Gamma}(\omega)$ is the coherence matrix composed of coherence functions, i.e.,

$$\mathbf{\Gamma}(\omega) = \begin{bmatrix} 1 & \gamma_{12}(\omega) & \cdots & \gamma_{1n}(\omega) \\ \gamma_{21}(\omega) & 1 & \cdots & \gamma_{2n}(\omega) \\ \vdots & \vdots & \ddots & \vdots \\ \gamma_{n1}(\omega) & \gamma_{n2}(\omega) & \cdots & 1 \end{bmatrix} \tag{6}$$

Because $\mathbf{\Gamma}(\omega)$ is also a Hermitian matrix with positive definite property, it can be decomposed by the following Cholesky decomposition

$$\mathbf{\Gamma}(\omega) = \mathbf{B}(\omega)\mathbf{B}^{T*}(\omega) \tag{7}$$

$$\mathbf{B}(\omega) = \begin{bmatrix} \beta_{11}(\omega) & 0 & \cdots & 0 \\ \beta_{21}(\omega) & \beta_{22}(\omega) & \cdots & 0 \\ \vdots & \vdots & \ddots & \vdots \\ \beta_{n1}(\omega) & \beta_{n2}(\omega) & \cdots & \beta_{nn}(\omega) \end{bmatrix} \tag{8}$$

where $*$ denotes complex conjugate; $\mathbf{B}(\omega)$ = decomposed coherence matrix and a lower triangular matrix; $\beta_{jj}(\omega), j = 1, 2, \cdots, n$ = a real and non-negative function of $\omega$; $\beta_{jk}(\omega), j = 1, 2, \cdots, n; j \neq k$ = a complex function of $\omega$; these complex functions can be written in polar form, i.e.,

$$\beta_{jk}(\omega) = \left|\beta_{jk}(\omega)\right|e^{i\theta_{jk}(\omega)} \tag{9}$$

where $\theta_{jk}(\omega)$ = complex phase of $\beta_{jk}(\omega)$.

Then, the samples of $x_j(t)$ can be generated by [35]

$$x_j(t) = 2\sqrt{\Delta\omega}\sum_{k=1}^{j}\sum_{l=1}^{N}\sqrt{S_{jj}(\omega_l)}\left|\beta_{jk}(\omega_l)\right|\cos[\omega_l t - \theta_{jk}(\omega_l) + \phi_{kl}]; j = 1, 2, \cdots, n \tag{10}$$

where $\omega_l = l\Delta\omega$ = discrete frequency; $N$ = frequency discrete number; $\Delta\omega = \omega_u/N$ = frequency increment; $\omega_u$ = upper cutoff frequency; $\phi_{kl}$ = independent random phase angle uniformly distributed in $[0, 2\pi]$. Obviously, the deterministic phase $\theta_{jk}(\omega)$ in the simulation Equation (10) is derived from the phase of coherence functions, reflecting the wave passage effect of the wind field.

In order to simulate the ergodic stationary wind field, the single-index frequency $\omega_l$ in Equation (10) need to be replaced by the double-index frequency $\omega_{kl}$, which can be calculated by [6]

$$\omega_{kl} = (l-1)\Delta\omega + \frac{k}{n}\Delta\omega, k = 1, 2, \cdots, n; l = 1, \cdots, N \tag{11}$$

## *2.2. Nonstationary Wind Fields*

The nonstationary wind field is generally characterized by the evolutionary power spectrum. Replace the PSD in stationary wind field simulation Equation (10) with the nonstationary evolutionary power spectrum density (EPSD), and then the stationary simulation algorithm can be extended to a non-stationary one, i.e., [9]

$$\hat{x}_j(t) = 2\sqrt{\Delta\omega} \sum_{k=1}^{j} \sum_{l=1}^{N} \sqrt{S_{jj}(\omega_l, t)} \left| \beta_{jk}(\omega_l) \right| \cos[\omega_l t - \theta_{jk}(\omega_l) + \phi_{kl}]; j = 1, 2, \cdots, n \quad (12)$$

where $\hat{x}_j(t)$ = the $j$th nonstationary wind fluctuation; $S_{jj}(\omega, t), j = 1, 2, \cdots, n$ = auto-EPSD of $\hat{x}_j(t)$.

Obviously, the above simulation Equation (12) is only suitable for the nonstationary wind fields with time-invariant coherence functions. When the wind field to be simulated has a time-varying coherence function $\gamma_{jk}(\omega, t)$, the Cholesky decomposition needs to be executed for the time-varying coherence matrix $\Gamma(\omega, t)$, and the corresponding simulation formulation is given as [27]

$$\hat{x}_j(t) = 2\sqrt{\Delta\omega} \sum_{k=1}^{j} \sum_{l=1}^{N} \sqrt{S_{jj}(\omega_l, t)} \left| \beta_{jk}(\omega_l, t) \right| \cos[\omega_l t - \theta_{jk}(\omega_l, t) + \phi_{kl}]; j = 1, 2, \cdots, n \quad (13)$$

where $\beta_{jk}(\omega, t)$ = the element in the decomposed time-varying coherence matrix; $\theta_{jk}(\omega, t)$ = complex phase of $\beta_{jk}(\omega, t)$.

## 3. Treatment of Phases of Coherence Functions

The deterministic phase angle in the simulation formula can be obtained in two ways: complex phase angle formula and separated phase. Here are two ways to solve the phase angle. Since the calculation of time-varying phase angle in time-varying coherent nonstationary simulation is similar, only the calculation of time-invariant phase angle is given.

### *3.1. Phase Formulation*

Based on the decomposition result $\beta_{jk}(\omega)$ of complex coherence matrix, its phase angle $\theta_{jk}(\omega)$ can be obtained by the phase formula of complex number and can be represented as

$$\theta_{jk}(\omega) = \begin{cases} \tan^{-1}\left\{ \frac{\mathrm{Im}[\beta_{jk}(\omega)]}{\mathrm{Re}[\beta_{jk}(\omega)]} \right\}, & \text{when } \mathrm{Re}[\beta_{jk}(\omega)] > 0 \\ \pm\frac{\pi}{2}, & \text{when } \mathrm{Re}[\beta_{jk}(\omega)] = 0 \text{ and } \mathrm{Im}[\beta_{jk}(\omega)] \neq 0 \\ \tan^{-1}\left\{ \frac{\mathrm{Im}[\beta_{jk}(\omega)]}{\mathrm{Re}[\beta_{jk}(\omega)]} \right\} \pm \pi, & \text{when } \mathrm{Re}[\beta_{jk}(\omega)] < 0 \text{ and } \mathrm{Im}[\beta_{jk}(\omega)] \neq 0 \\ \pi, & \text{when } \mathrm{Re}[\beta_{jk}(\omega)] < 0 \text{ and } \mathrm{Im}[\beta_{jk}(\omega)] = 0 \end{cases} \quad (14)$$

where "Im" and "Re" denote imaginary and real parts, respectively. Based on the definition of complex phase, its value should fall into the range of $[-\pi, \pi]$. However, in almost all previous studies [6], the phase is generally expressed as

$$\theta_{jk}(\omega) = \tan^{-1}\left\{ \frac{\mathrm{Im}\left[\beta_{jk}(\omega)\right]}{\mathrm{Re}\left[\beta_{jk}(\omega)\right]} \right\} \quad (15)$$

In this study, this is termed as "classical phase formula". Obviously, Equation (15) can only represent the phase in the range of $(-\pi/2, \pi/2)$. Thus, the complex phase calculated by Equation (15) may introduce errors for the simulation results. In this study, the influence of Equation (15) on the four wind field simulations will be studied in detail, in terms of time history samples and correlation functions.

*3.2. Separated Phase*

On the other hand, the phases in coherence matrix can be separated out to avoid participating in the Cholesky decomposition [36]. The coherence matrix $\mathbf{\Gamma}(\omega)$ is first represented by its modulus matrix and phase matrix, i.e.,

$$\mathbf{\Gamma}(\omega) = \mathbf{\Pi}(\omega) \otimes \mathbf{\Theta}(\omega) \tag{16}$$

where $\otimes$ denotes the multiplication operation between the corresponding elements of two matrices; $\mathbf{\Pi}(\omega)$ = lagged coherence matrix and given by

$$\mathbf{\Pi}(\omega) = \begin{bmatrix} 1 & |\gamma_{12}(\omega)| & \cdots & |\gamma_{1n}(\omega)| \\ |\gamma_{21}(\omega)| & 1 & \cdots & |\gamma_{2n}(\omega)| \\ \vdots & \vdots & \ddots & \vdots \\ |\gamma_{n1}(\omega)| & |\gamma_{n2}(\omega)| & \cdots & 1 \end{bmatrix} \tag{17}$$

and $\mathbf{\Theta}(\omega)$ = phase matrix of $\mathbf{\Gamma}(\omega)$ and given by

$$\mathbf{\Theta}(\omega) = \begin{bmatrix} 1 & e^{i\vartheta_{12}(\omega)} & \cdots & e^{i\vartheta_{1n}(\omega)} \\ e^{i\vartheta_{21}(\omega)} & 1 & \cdots & e^{i\vartheta_{2n}(\omega)} \\ \vdots & \vdots & \ddots & \vdots \\ e^{i\vartheta_{n1}(\omega)} & e^{i\vartheta_{n2}(\omega)} & \cdots & 1 \end{bmatrix} \tag{18}$$

Because $\mathbf{\Pi}(\omega)$ is also a Hermitian matrix with positive definite property, it can also be decomposed by the Cholesky decomposition, i.e.,

$$\mathbf{\Pi}(\omega) = \mathbf{L}(\omega)\mathbf{L}^T(\omega) \tag{19}$$

where $\mathbf{L}(\omega)$ = the decomposed lower triangular matrix. According to [36], the following two equations are true:

$$\mathbf{L}(\omega) = \begin{bmatrix} |\beta_{11}(\omega)| & 0 & \cdots & 0 \\ |\beta_{21}(\omega)| & |\beta_{22}(\omega)| & \cdots & 0 \\ \vdots & \vdots & \ddots & \vdots \\ |\beta_{n1}(\omega)| & |\beta_{n2}(\omega)| & \cdots & |\beta_{nn}(\omega)| \end{bmatrix} \tag{20}$$

and

$$\theta_{jk}(\omega) = \vartheta_{jk}(\omega) \tag{21}$$

## 4. Verification of Correlation Functions

The simulation effectiveness is generally verified by comparing the consistency of the simulated and target correlation functions. In the following, the influence of phase on correlation functions for four wind field simulations will be respectively discussed.

*4.1. Stationary Process*

Based on the target PSD and coherence function, the target auto-/cross- correlation functions $R_{jk}^0(\tau)$ of the stationary process can be calculated by

$$R_{jk}^0(\tau) = \int_{-\infty}^{\infty} \sqrt{S_{jj}(\omega)S_{kk}(\omega)}\gamma_{jk}(\omega)e^{i\omega\tau}d\omega; j, k = 1, 2, \cdots, n \tag{22}$$

In order to verify the effectiveness of the simulated nonergodic wind field, the ensemble auto-/cross-correlation functions $R_{jk}(\tau)$ need to be calculated by the generated multiple numerical samples $x_j(t)$, $j = 1, 2, \cdots, n$, i.e.,

$$R_{jk}(\tau) = E[x_j(t)x_k(t + \tau)] \tag{23}$$

where $E[\cdots]$ denotes mathematical expectation. For the ergodic wind field, the temporal auto-/cross-correlation function $R_{jk}^{(i)}(\tau)$ need to be calculated by the generated numerical samples $x_j(t)$, $j = 1, 2, \cdots, n$, i.e.,

$$R_{jk}^{(i)}(\tau) = \frac{1}{T}\int_0^T x_j(t)x_k(t + \tau)dt \tag{24}$$

where $T$ = time duration of ergodic samples. When the estimated correlation functions match the corresponding targets, the simulated wind field is considered to be effective.

Before numerical examples, it is necessary to qualitatively study the influence of phase on simulation from the perspective of theoretical analysis, as follows.

Based on the stationary simulation formulation Equation (10), the ensemble auto-/cross-correlation function of the simulated samples $x_j(t)$, $j = 1, 2, \cdots, n$ can be represented by

$$R_{jk}(\tau) = 2\Delta\omega \sum_{m=1}^{n} \sum_{l=1}^{N} \sqrt{S_{jj}(\omega_l)S_{kk}(\omega_l)}|\beta_{jm}(\omega_l)||\beta_{km}(\omega_l)| \\ \times \cos[\omega_l\tau + \theta_{jm}(\omega_l) - \theta_{km}(\omega_l)]; j, k = 1, 2, \cdots, n \tag{25}$$

which will converge to the corresponding target $R_{jk}^0(\tau)$ as the frequency increment $\Delta\omega$ decreases. Equation (25) presents the analytical correlation functions, which are theoretically identical to the numerical simulation results. Therefore, they can be utilized to qualitatively analyze the influence of phases on correlation functions. It can be seen from Equation (25) that the ensemble autocorrelation function of the simulated samples is independent on the phase, while the ensemble cross-correlation function of the simulated samples is related to the phase.

For the ergodic stationary simulation, the single-index $\omega_l$ in Equation (25) is replaced by the double-index $\omega_{ml}$, and the ensemble auto-/cross-correlation function of the simulated samples $x_j(t)$, $j = 1, 2, \cdots, n$ can be expressed as [6]

$$R_{jk}(\tau) = 2\Delta\omega \sum_{m=1}^{n} \sum_{l=1}^{N} \sqrt{S_{jj}(\omega_{ml})S_{kk}(\omega_{ml})}|\beta_{jm}(\omega_{ml})||\beta_{km}(\omega_{ml})| \\ \times \cos[\omega_{ml}\tau + \theta_{jm}(\omega_{ml}) - \theta_{km}(\omega_{ml})]; j, k = 1, 2, \cdots, n \tag{26}$$

When the time duration of the simulated sample $T$ is equal to the period $T_0$, the temporal auto-/cross-correlation function $R_{jk}^{(i)}(\tau)$ of the sample function $x_j(t)$, $j = 1, 2, \cdots, n$ has the same expression, i.e.,

$$R_{jk}^{(i)}(\tau) = R_{jk}(\tau) \text{ for } T = T_0 \tag{27}$$

$$T_0 = n\frac{2\pi}{\Delta\omega} \tag{28}$$

Obviously, the temporal autocorrelation function of the simulated ergodic samples is also independent on the phase, while the temporal cross-correlation function is related to the phase.

### 4.2. Nonstationary Process

Based on the target EPSD and coherence function, the target auto-/cross- correlation functions $R_{jk}^0(t, t + \tau)$ of the time-invariant coherent nonstationary process can be calculated by

$$R_{jk}^0(t, t + \tau) = \int_{-\infty}^{\infty} \sqrt{S_{jj}(\omega, t)S_{kk}(\omega, t + \tau)}\gamma_{jk}(\omega)e^{i\omega\tau}d\omega; j, k = 1, 2, \cdots, n \tag{29}$$

In order to verify the effectiveness of the simulated nonstationary wind field, the ensemble auto-/cross-correlation functions $R_{jk}(t, t + \tau)$ need to be calculated by the generated multiple numerical samples $\hat{x}_j(t)$, $j = 1, 2, \cdots, n$, i.e.,

$$R_{jk}(t, t + \tau) = E[\hat{x}_j(t)\hat{x}_k(t + \tau)] \tag{30}$$

Similar to the stationary simulation, the analytical correlation functions of the nonstationary simulation can be also obtained by its simulation formula, i.e., Equation (12). The ensemble auto-/cross-correlation function of the simulated nonstationary samples $\hat{x}_j(t)$, $j = 1, 2, \cdots, n$ can be represented by [9]

$$\begin{aligned} R_{jk}(t, t + \tau) &= 2\Delta\omega \sum_{m=1}^{n} \sum_{l=1}^{N} \sqrt{S_{jj}(\omega_l, t)S_{kk}(\omega_l, t + \tau)} |\beta_{jm}(\omega_l)||\beta_{km}(\omega_l)| \\ &\times \cos[\omega_l\tau + \theta_{jm}(\omega_l) - \theta_{km}(\omega_l)]; j, k = 1, 2, \cdots, n \end{aligned} \tag{31}$$

It can be seen that the ensemble autocorrelation function is independent of the phase, while the ensemble cross-correlation function will be affected by the phase.

For the time-varying coherent nonstationary simulation, the target auto-/cross- correlation functions $R_{jk}^0(t, t + \tau)$ can be computed by [27]

$$\begin{aligned} R_{jk}^0(t, t + \tau) &= \int_{-\infty}^{\infty} \sqrt{S_{jj}(\omega, t)S_{kk}(\omega, t + \tau)} \sum_{m=1}^{n} \beta_{jm}(\omega, t)\beta_{km}^*(\omega, t + \tau)e^{i\omega\tau}d\omega; \\ &j, k = 1, 2, \cdots, n \end{aligned} \tag{32}$$

Based on Equation (13), the ensemble auto-/cross-correlation function of the simulated time-varying coherent nonstationary samples $\hat{x}_j(t)$, $j = 1, 2, \cdots, n$ can be represented by

$$\begin{aligned} R_{jk}(t, t + \tau) &= 2\Delta\omega \sum_{m=1}^{n} \sum_{l=1}^{N} \sqrt{S_{jj}(\omega_l, t)S_{kk}(\omega_l, t + \tau)} |\beta_{jm}(\omega_l, t)||\beta_{km}(\omega_l, t + \tau)| \\ &\times \cos[\omega_l\tau + \theta_{jm}(\omega_l, t) - \theta_{km}(\omega_l, t + \tau)]; j, k = 1, 2, \cdots, n \end{aligned} \tag{33}$$

Clearly, both the ensemble auto- and cross-correlation functions are dependent on the phase.

## 5. Numerical Examples

In this section, the influence of the phase on the simulated samples and correlation functions will be quantitatively discussed by the four numerical examples corresponding to four wind fields.

It is assumed that the four wind fields possess the same simulation points distributing along the deck of a cable-stayed bridge with the length of 400 m. These wind fields consist of 41 simulation points, where the distance between two adjacent simulation points is 10 m, as indicated in Figure 1.

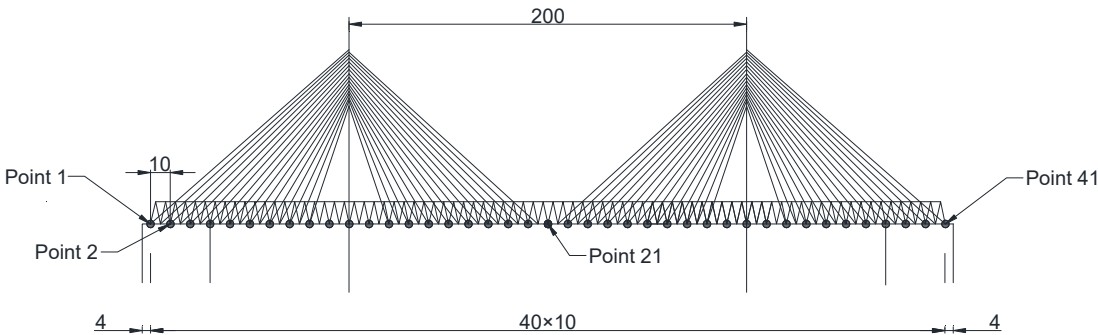

**Figure 1.** Simulation points distributed on bridge deck.

*5.1. Stationary Wind Fields*

In terms of stationary wind field, the widely used two-sided Kaimal spectrum is selected as the target PSD, i.e.,

$$S(\omega) = \frac{200}{4\pi} \frac{z}{U} \frac{u_*^2}{\left[1 + 50\frac{\omega z}{2\pi U}\right]^{\frac{5}{3}}} \tag{34}$$

where the height of simulation points $z = 60$ m; the mean wind velocity $U = 40$ m/s; the shear velocity $u_* = kU/\ln(z/z_0)$ in which the coefficient $k = 0.4$ and the ground roughness $z_0 = 0.01$ m. It is assumed that all the simulation points possess the same auto-PSDs.

The coherence function between two simulation points adopts the following Davenport's coherence function with the complex phase [37]

$$\gamma_{jk}(\omega) = \exp\left(-\frac{C_y|y_j - y_k|\omega}{2\pi U}\right) \exp\left(-i\frac{(y_j - y_k)\omega}{v_{app}}\right) \tag{35}$$

where $y_j$ and $y_k$ = the horizontal coordinates of Points $j$ and $k$; $C_y$ = attenuation coefficient and takes 10; $v_{app}$ = apparent wave velocity and takes 10 m/s. Then, the deterministic phase $\theta_{jk}(\omega)$ in simulation formulation can be readily calculated by the aforementioned two schemes, based on the given coherence function. The typical phase results $\theta_{42}(\omega)$ obtained by the classical formula and separated phase are, respectively, shown in Figure 2. It can be seen that the two phases are periodic and the period of the latter is twice that of the former. Additionally, the value range of the latter is twice that of the former.

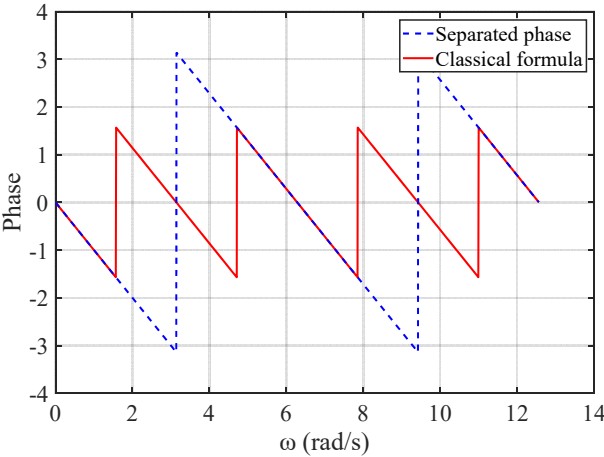

**Figure 2.** Phases obtained by two schemes.

5.1.1. Non-Ergodic Stationary Wind Fields

In the non-ergodic stationary wind field, the time and frequency parameters are selected as: upper cutoff frequency $\omega_u = 4\pi$ rad/s; frequency interval $\Delta\omega = 0.006$ rad/s; number of frequency divisions $N = 2048$; time duration 1024 s; time interval $\Delta t = 0.25$ s; number of time divisions $M = 4096$.

Figure 3 shows the simulated sample time histories at Points 1, 21, and 41, based on the above two phase results under the same random phase. It can be seen that the two results at Point 1 are identical and have no difference, while the results at Points 21 and 41 have an evident distinction and the difference becomes larger and larger with the increase in the distance. This is because the sample at Point $j$ is only related to the $j$th row of the decomposed coherence matrix, as shown in Equation (10). The larger the simulation point number, the more complex phases involved in the simulation, and the greater the influence of the phase on the simulation results. Therefore, using the classical formula to calculate the phase will have a certain impact on the simulated samples, and this impact will be greater for the later simulation points.

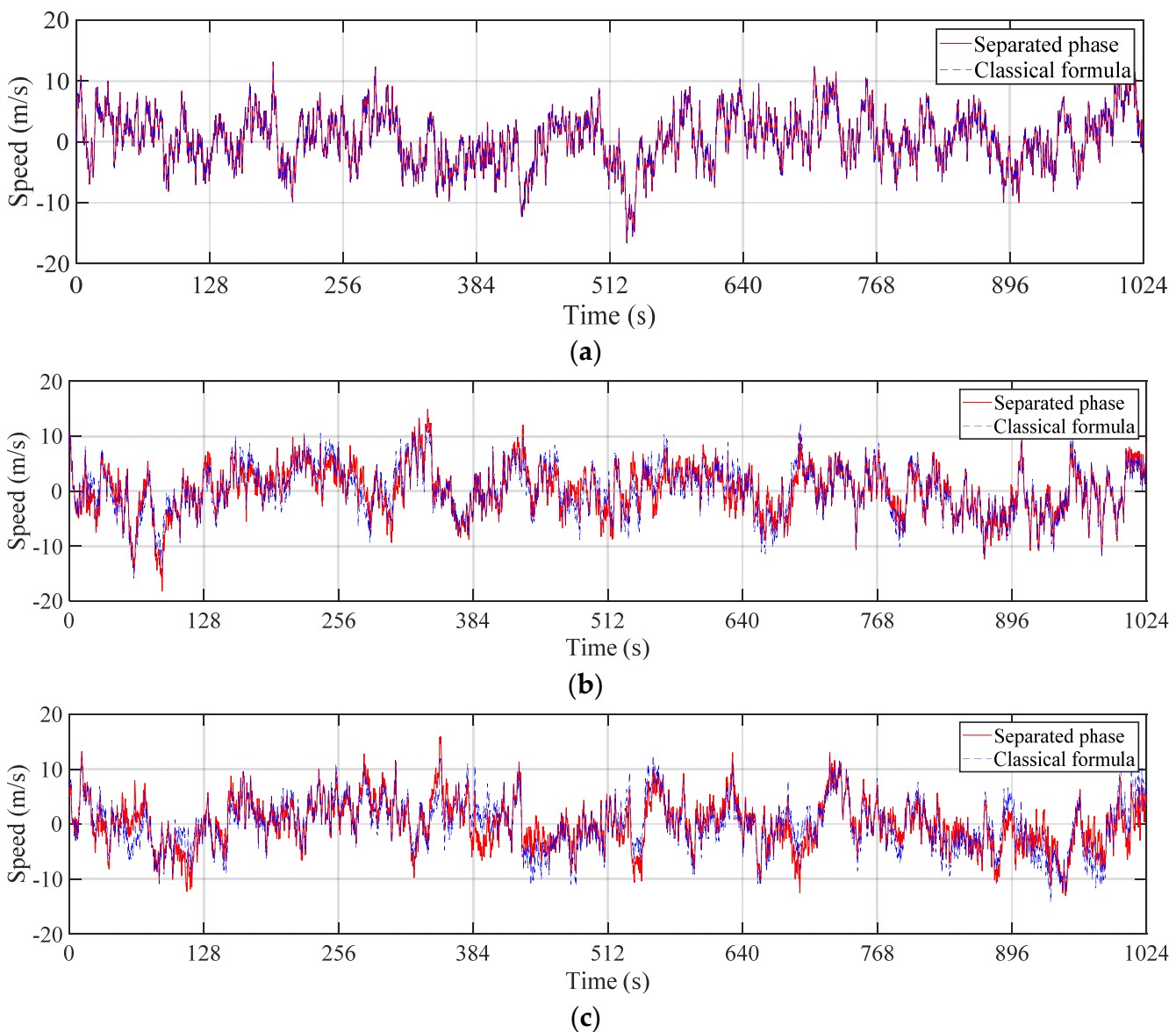

**Figure 3.** Simulated nonergodic stationary wind fluctuation samples: (**a**) Point 1; (**b**) Point 21; (**c**) Point 41.

　　　In order to further study the influence of the phase on correlation function, 1000 samples are generated to estimate the ensemble auto-/cross- correlation function by Equation (23). Figure 4 shows a typical autocorrelation function $R_{21,21}(\tau)$ of Point 21 and a typical cross-correlation function $R_{1,21}(\tau)$ between Points 1 and 21, where the targets are calculated by Equation (22). Comparing the results based on two phase schemes and the corresponding target, it can be found that two estimated autocorrelation functions are completely coincident, which shows that the phase has no effect on the autocorrelation function. However, the estimated cross-correlation functions are totally different, and the result obtained by the classical phase formula seriously deviated from the target. Therefore, the influence of the phase on the cross-correlation function is considerably great and the classical phase formula will cause the obvious simulation error.

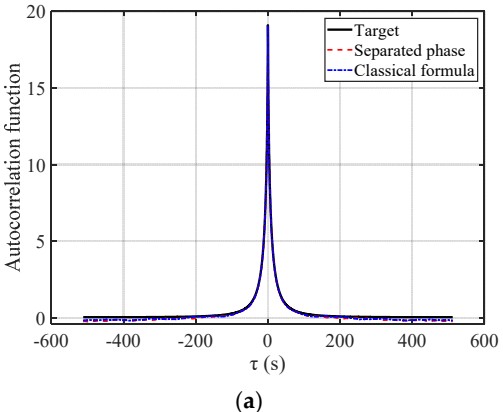

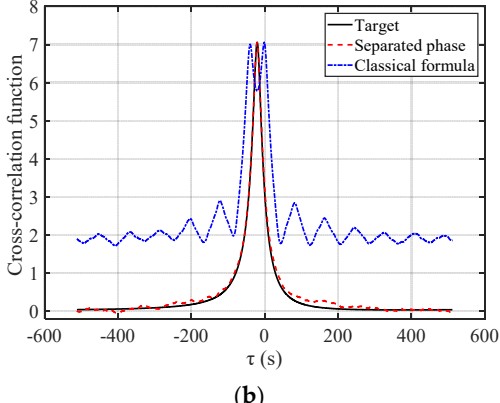

(**a**)                        (**b**)

**Figure 4.** Estimated correlation functions by nonergodic stationary wind fluctuation samples (1000 samples): (**a**) $R_{21,21}(\tau)$; (**b**) $R_{1,21}(\tau)$.

### 5.1.2. Ergodic Stationary Wind Fields

Based on the double-index frequency, $\omega_{kl}$, the ergodic stationary wind field can be simulated. The time duration of generated samples adopts a period, i.e., 41,984 s. The other time and frequency parameters are the same as the last example. The first 1024 s of generated ergodic wind fluctuation samples at Points 1, 21, and 41 are displayed in Figure 5, which comparing the samples obtained by two phase schemes under the same random phase. Obviously, the influence of the phase on the simulated ergodic sample is similar to that of the non-ergodic stationary simulation. Therefore, using the classical phase formula to simulate the ergodic stationary wind field will still cause errors.

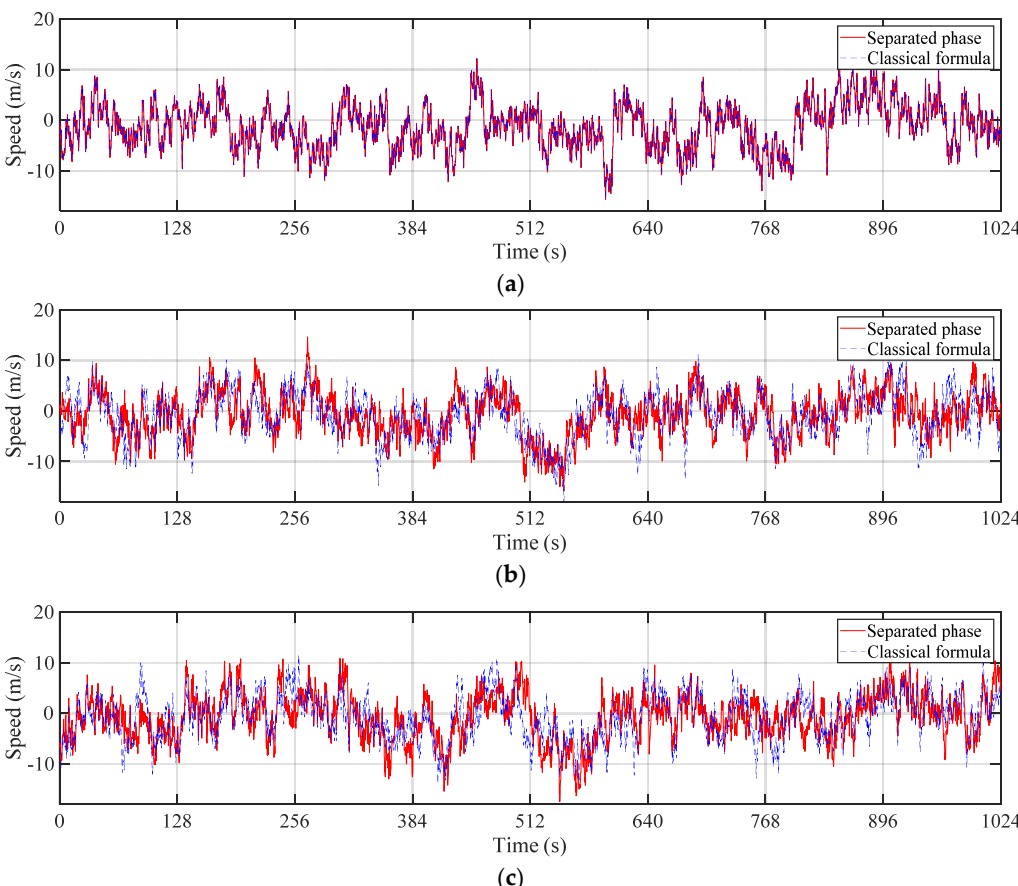

**Figure 5.** Simulated ergodic stationary wind fluctuation samples (first 1024 s): (**a**) Point 1; (**b**) Point 21; (**c**) Point 41.

Based on Equation (24), the temporal auto-/cross-correlation functions of the simulated ergodic samples can be estimated. Figure 6 shows the comparison of two typical correlation functions obtained by two phase schemes with the corresponding targets, where the targets are also calculated by Equation (22). The same phenomenon as the above non-ergodic simulation can be observed in this ergodic simulation. Thus, the classical phase formula will also lead to the incorrect simulation of ergodic wind fields.

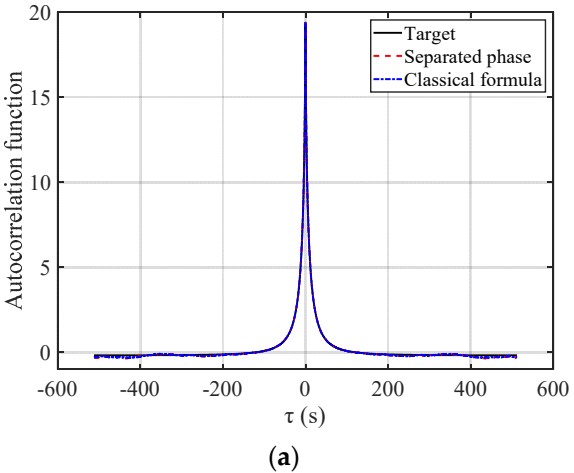

(a)

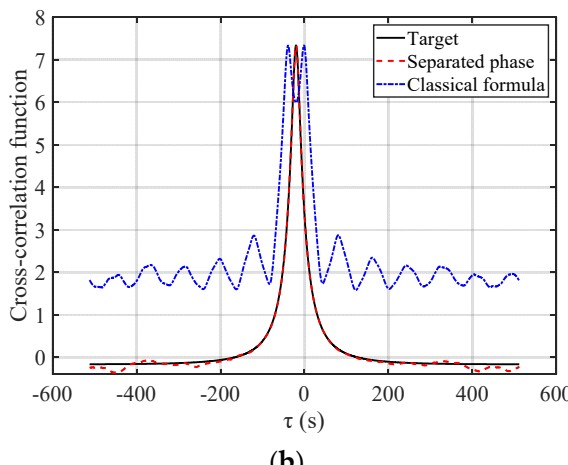

(b)

**Figure 6.** Estimated correlation functions by ergodic stationary wind fluctuation samples: (a) $R_{21,21}(\tau)$; (b) $R_{1,21}(\tau)$.

### 5.2. Nonstationary Wind Fields

For the nonstationary wind field, the target auto-EPSD is assumed to be the extended Kaimal spectrum, which can be obtained by replacing the constant mean wind velocity as the time-varying one $\hat{U}(t)$ in Equation (34). The time-varying mean wind velocity $\hat{U}(t)$ is generally given by

$$\hat{U}(t) = d(t)U \tag{36}$$

where $d(t)$ = time modulation function with three parameters [38], i.e.,

$$d(t) = \alpha_0 t^{\beta_0} e^{-\lambda t}, \alpha_0 > 0, \beta_0, \lambda \geq 0 \tag{37}$$

in which the parameter $\alpha_0 = \lambda^{\beta_0}/\beta_0^{\beta_0} e^{\beta_0}$ and assume the maximum value of this function occurs at $t_{\max} = \beta_0/\lambda$. In this simulation, the parameters $t_{\max} = 300$ and $\beta_0 = 2$ are adopted. Assume that each simulation point has the same auto-EPSD.

The time-invariant coherence function adopts Equation (35). The time-varying coherence function is obtained by extending the Davenport's model, i.e., [37]

$$\gamma_{jk}(\omega, t) = \exp\left(-\frac{C_y|y_j - y_k|\omega}{2\pi U(t)}\right) \exp\left(-i\frac{(y_j - y_k)\omega}{v_{app}^{jk}(t)}\right) \tag{38}$$

where $U(t) = [U_j(t) + U_k(t)]/2$ = averaged time-varying mean wind velocity and adopts the following simplified form:

$$U(t) = U \cdot [1 + \gamma \cos(\omega' t)] = 40[1 + 0.3\cos(2\pi/1200 \cdot t)] \tag{39}$$

and the apparent wave velocity $v_{app}^{jk}(t)$ is given by [37]

$$v_{app}^{jk}(t) = \frac{2\pi U(t)}{C_\theta} \tag{40}$$

in which $C_\theta$ = a coefficient obtained by experiments and generally takes 5.5.

In the following, two nonstationary wind fields with time-invariant coherence and time-varying coherence will be simulated based on the two phase schemes to study the influence of the phase on the nonstationary simulation. The time and frequency parameters used to the nonstationary wind field are the same as the above non-ergodic stationary simulation.

### 5.2.1. Nonstationary Wind Fields with Time-Invariant Coherence

Figure 7 presents the comparison of two samples obtained by two phase schemes with the same random phase at Points 1, 21, and 41. It can be observed that the samples at Point 1 are totally consistent and the samples at other simulation points have more or less differences. Therefore, the nonstationary simulation using the classical phase formula possesses an evident error. Furthermore, the error becomes larger and larger as the simulation point number increases.

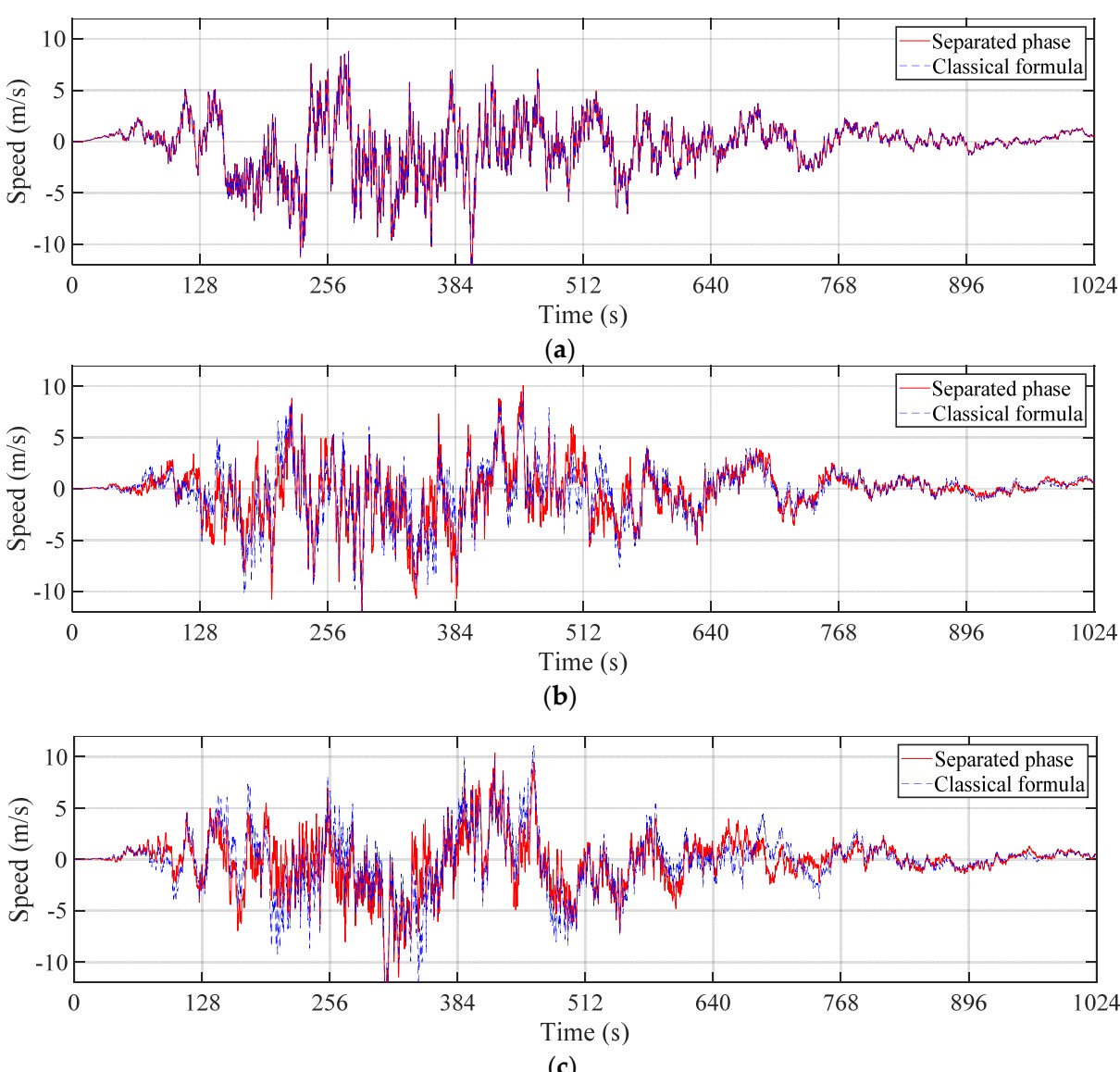

**Figure 7.** Simulated time-invariant coherent nonstationary wind fluctuation samples: (**a**) Point 1; (**b**) Point 21; (**c**) Point 41.

Further, 1000 samples are generated to estimate the ensemble auto-/cross-correlation functions based on Equation (30). The target correlation functions are computed by Equation (29). Figure 8 compares the typical correlation functions obtained by the two phase

schemes with the corresponding targets when the time lag $\tau = 0$, 16, and 32 s. It can also be found that the phase has no effect on autocorrelation function, while the influence on the cross-correlation function is rather strong. This shows that the nonstationary simulation using the classical phase formulation is erroneous.

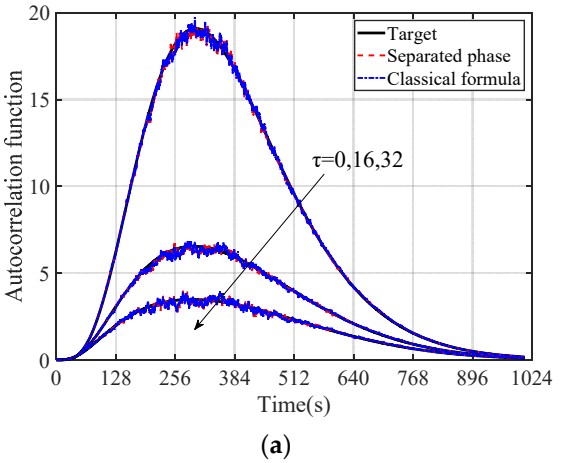
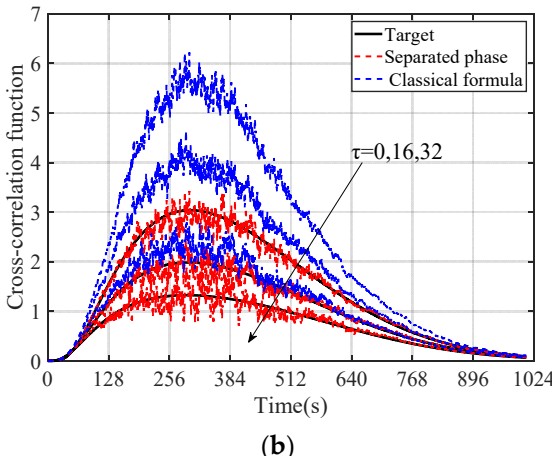

**(a)**          **(b)**

**Figure 8.** Estimated correlation functions by time-invariant coherent nonstationary wind fluctuation samples (1000 samples): (**a**) $R_{21,21}(t, \tau)$; (**b**) $R_{1,21}(t, \tau)$.

### 5.2.2. Nonstationary Wind Fields with Time-Varying Coherence

Based on the given time-varying coherence function, the deterministic phase $\theta_{jk}(\omega, t)$ in the simulation formulation can be calculated by the aforementioned two schemes. The typical time-varying phases $\theta_{12}(\omega, t)$ obtained by two schemes are shown in Figure 9. Obviously, two phases are the periodic function of $\omega$. The amplitude and period of the separated phase are twice the phase, using the classical formulation.

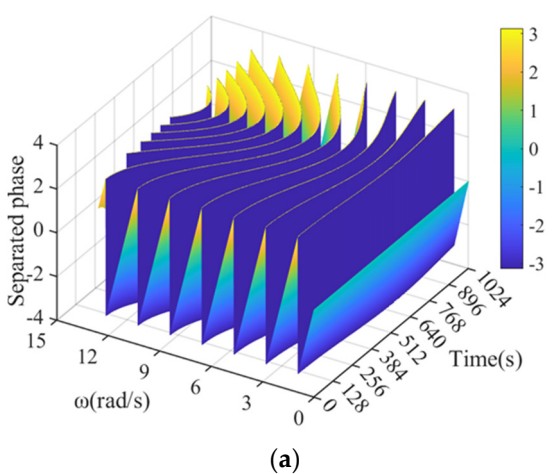
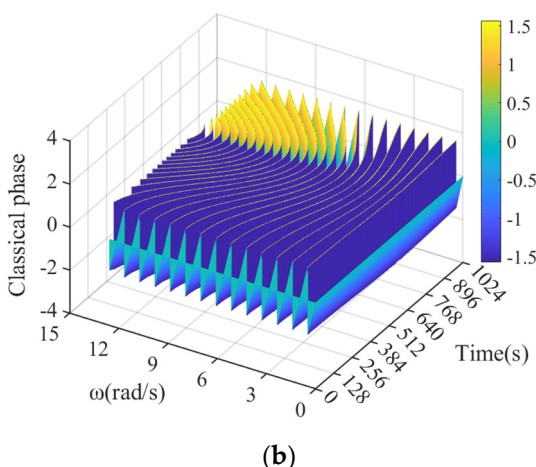

**(a)**          **(b)**

**Figure 9.** Time-varying phases obtained by two schemes: (**a**) separated phase; (**b**) classical formulation.

Substituting the phases to the simulation formulation, the time-varying coherent nonstationary wind fluctuation samples can be generated. Figure 10 displays three typical sample time histories and compares the samples based on the above two phases. The similar laws can be observed. The higher the phase participation of the sample, the greater the error caused by the classical phase formula. In addition, it can be also found that the influence of the phase on the time-varying coherent nonstationary simulation is less than the time-invariant coherent nonstationary simulation.

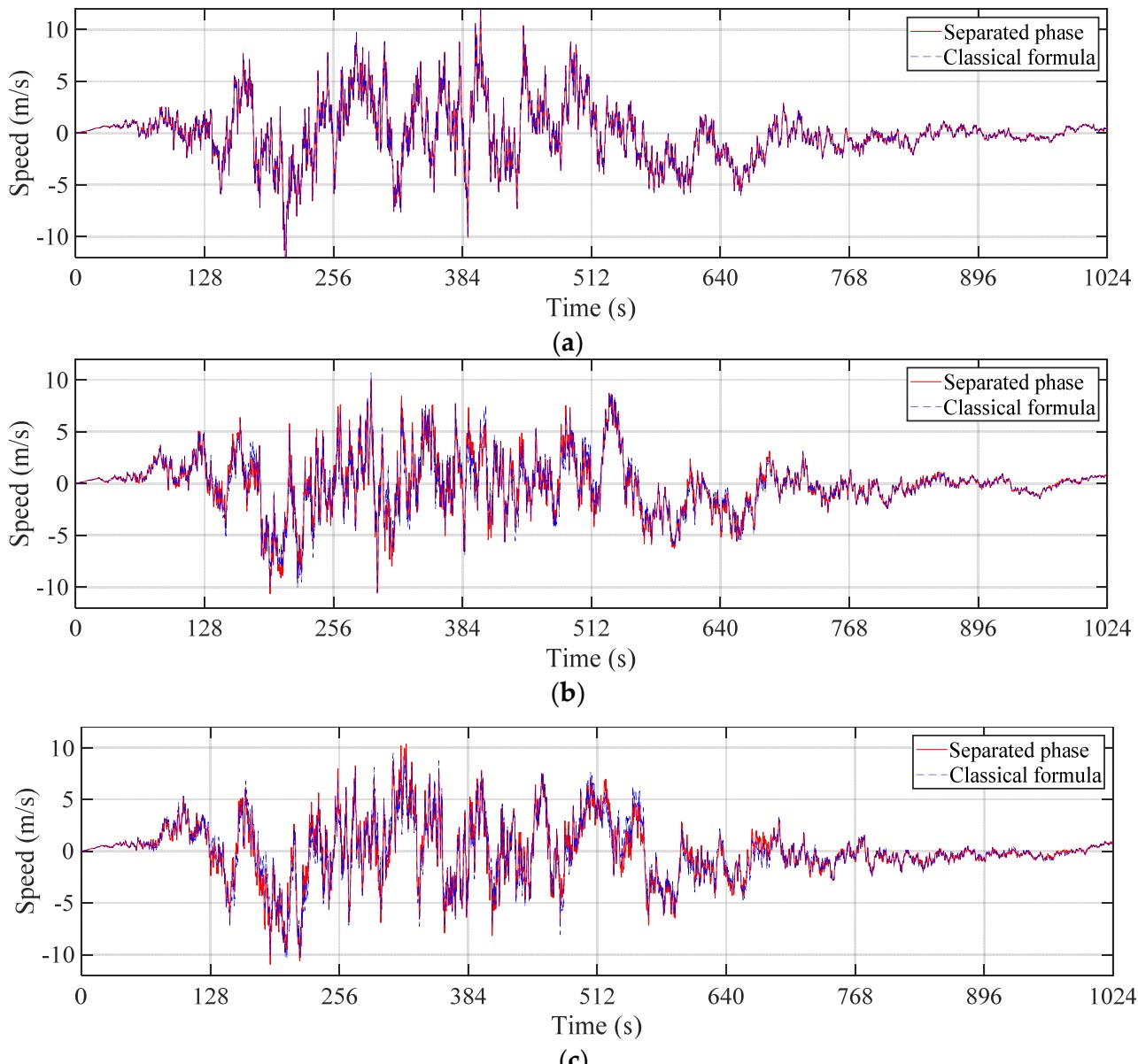

**Figure 10.** Simulated time-varying coherent nonstationary wind fluctuation samples: (**a**) Point 1; (**b**) Point 21; (**c**) Point 41.

The ensemble auto-/cross-correlation functions can be estimated by 1000 simulated samples, based on Equation (30). The target correlation functions are calculated by Equation (32). Figure 11 verifies the estimated correlation functions and the targets when the time lag $\tau = 0$, 16, and 32 s. Although the autocorrelation function is affected by the phase (see Equation (33)), the impact is small and can be ignored (see Figure 11). However, the cross-correlation function based on the classical phase formula is still seriously deviated from the target. Therefore, the classical phase formula cannot be directly used in the nonstationary simulation.

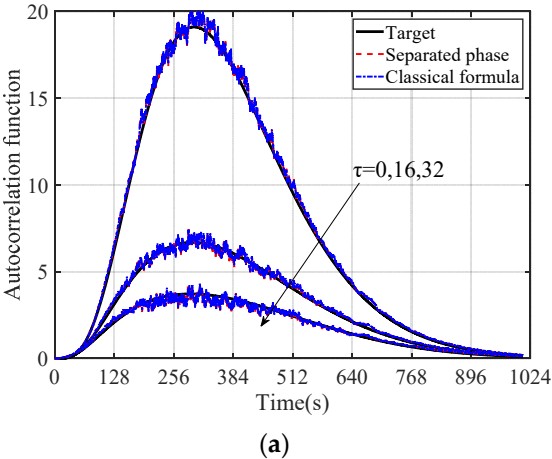
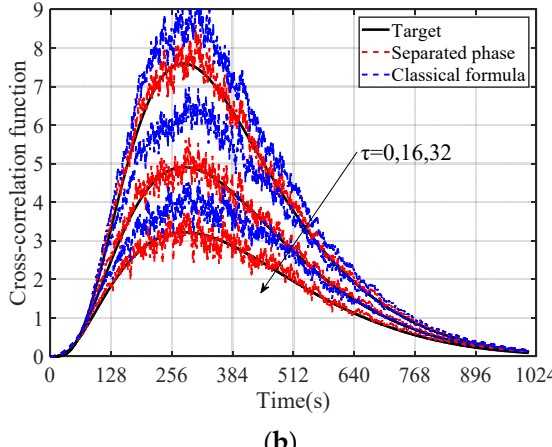

(**a**)                                      (**b**)

**Figure 11.** Estimated correlation functions by time-varying coherent nonstationary wind fluctuation samples (1000 samples): (**a**) $R_{21,21}(t,\tau)$; (**b**) $R_{1,21}(t,\tau)$.

## 6. Conclusions

The wave passage effect of multi-point wind field is generally considered by the phase of coherence function. However, in the wind field simulation by the spectral representation method, the classical phase formula is not rigorous. This will affect the accuracy of the simulation results and even cause incorrect simulations. In this study, the influences of the phase on four wind field simulations are investigated and discussed in detail. The qualitative analysis based on the theoretical correlation function formula is first made to study the influence of the phase. Then, four numerical examples are utilized to quantitatively study the magnitude of the influence on the sample time history and correlation function of the simulated wind field. Some conclusions are given as follows:

1.  The period of the classical phase is only 50% of the exact result and the amplitude of the classical phase is also only 50% of the exact result.
2.  The errors induced by the phase formula occur mainly in the simulation results with high phase participation, such as the samples at the farther simulation points and all cross-correlation functions. The maximum error is even more than 100%. The sample at the first simulation point and all autocorrelation functions are not affected by the phase formula.
3.  The classical phase formula has a serious influence on the simulation of four kinds of wind fields. The influence of the phase on the non-ergodic and ergodic stationary wind fields are roughly equivalent. The influence of the phase on the time-varying coherent nonstationary simulation is less than the time-invariant coherent nonstationary simulation.
4.  The classical phase formula cannot be used for the wind field simulation, considering the wave passage effect, otherwise it will result in incorrect simulation results and then affect the subsequent structural analysis.

**Author Contributions:** Conceptualization, N.Z., L.P. and X.W.; methodology, N.Z. and X.W.; software, X.L.; validation, Z.X.; formal analysis, X.C. and X.W.; writing—original draft preparation, N.Z. and X.L.; writing—review and editing, N.Z., X.L. and L.P.; supervision, N.Z. and X.W.; project administration, N.Z. and L.P. All authors have read and agreed to the published version of the manuscript.

**Funding:** This research was funded by the National Natural Science Foundation of China (Grant Nos. 52108464 and 51808078), 111 Project of China (Grant No. B18062), the CRSRI Open Research Program (Program SN: CKWV2019733/KY), the Dual Support Plan of Sichuan Agricultural University (Grant No. 2021993510), and the Research Funding for Supporting the Postdoctoral Working in Chongqing (Grant No. 2020LY06).

**Conflicts of Interest:** The authors declare no conflict of interest.

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
