# Peer review of "Influence of Phases of Coherence Functions on the Wind Field Simulation Using Spectral Representation Method"

_applsci, doi:10.3390/app12168190_

Round 1

Reviewer 1 Report

1. Since the method is very empirical and violates the fundamental fluid mechanics equations, I am not sure how much this method is applicable.

2. Since I don't have background on the particular work, I couldn't help you much.

Reviewer 2 Report

The reviewed manuscript concerns on influence of phases of coherence functions on the wind field simulation using spectral representation method. The article examines and discusses the phase effect on stationary and non-stationary wind field simulations. Two schemes containing a classical phase formula and a separate phase scheme were compared - for four types of wind field simulations. The aim of the research has been precisely defined and achieved. The subject of the manuscript is in line with the journal's profile. I propose to accept the article after taking one correction into account:

Conclusions should be supported with specific numerical values (e.g. expressed in%). Additionally, the authors should indicate more precisely the potential practical aspect of the obtained results. 

Reviewer 3 Report

The phase coherence functions on various wind field simulations are presented in this paper.

The paper is well presented however following points need a bit more clarification.

1) Please define what you mean by the wave in the introduction portion or abstract.

So that the reader new to this field can easily make the picture in his mind.

2) Could you please define how the time-invariant wind field can be non-stationary?

A little explanation of all four cases can improve the readability of the article.

3) Moreover, why only stationary and non-stationary cases are discussed only and the rest of the two are left?

Reviewer 4 Report

Eq.(4) might be messed up. Diagonal matrix should be in the middle with no need for transpose.

The paper is well written and ready for publication. 
